**EMBO**
**Molecular Medicine**

# Dominant Gα mutations in human disease: unifying mechanisms and treatment strategies

Vladimir L Katanaev [1,2] & Gonzalo P Solis [1]

## Abstract

Sixteen Gα-subunits transduce hundreds of G protein-coupled receptors and control countless cellular activities. Mutations in respective *GNA* genes underlie developmental, oncological, metabolic, neurological, and other pathologies. In addition to classical loss-of-function (LOF) and gain-of-function (GOF) mutations (the former represented by gene deletions/truncations, the latter by specific GTP hydrolysis-deficient mutations), multiple pathogenic dominant missense variants have been discovered in *GNA* genes, and their numbers constantly increase through advanced genetic diagnostics. While these mutations often have confusing features of hypomorphic, dominant-negative, and GOF mutations, many of the pathogenic Gαo (and by inference, other Gα-subunit) variants have recently emerged as neomorphic, i.e., leading to the creation of novel dominant pathogenic functions. Cross-family analysis of these missense variants scattered across *GNA* genes permits establishing mutational signatures underlying a wide range of Gα-pathies. These mutation patterns have a strong predictive power in the following aspects. First, new dominant mutations in further *GNA* genes will be discovered in rare diseases. Second, unifying mechanisms of pathogenic dominance emerge in different Gα-subunits. And third, drug(s) acting against some Gα-pathies may prove effective against others.

## Omnipresence of dominant—yet distinct from the classical activating —mutations across Gα-subunits in diverse human diseases

Heterotrimeric G proteins act as immediate transducers of G protein-coupled receptors (GPCRs) —the largest receptor family in metazoans, controlling innumerable developmental, physiological, and pathological processes, and the target of ca. 50% of all currently marketed drugs. Heterotrimeric G proteins consist of the α-, β-, and γ-subunits, of which the α-subunits determine the GPCR-coupling and signaling effector specificity. In humans, 16 genes encoding Gα-subunits exist, producing proteins that are grouped into four subclasses: Gαi/o, Gαs, Gαq, and Gα12/13 (Pierce et al, 2002).

When bound to GDP, the G protein can exist as a heterotrimer and is competent to interact with the cognate GPCR. The activated GPCR acts as a guanine nucleotide exchange factor (GEF), catalyzing the exchange of GDP for GTP on Gα-subunits. This triggers dissociation of the G protein into Gα-GTP and the Gβγ-heterodimer, both transmitting the signal further downstream. When GTP on Gα is hydrolyzed, the inactive Gαβγ heterotrimer re-associates for a new cycle of activation by the GPCR (Fig. 1). The GTP hydrolysis is strongly stimulated by a group of GAP (GTPase activating protein) regulators, most of which belong to the RGS (regulator of G protein signaling) family (Ross and Wilkie, 2000).

Gα-subunits are composed of three distinct domains: a Ras-like domain (RD), an adjacent α-helical domain (AHD), and the N-terminal α-helix (αN) that projects away from the other domains (Fig. 2A). Mutations in the RD that kill the GTPase activity of Gα-subunits render the proteins constitutively active and can cause cancer (Arang and Gutkind, 2020). Key amino acid motifs involved in guanine nucleotide binding and hydrolysis are distributed throughout the RD (Fig. 2A (Sprang, 1997)). The classical oncogenic mutations occur in the conserved Arg of the Switch I region (R179 in Gαo) or Gln of the Switch II (Q205 in Gαo) (boxed in Fig. 3); the former is exemplified by Gαi2 [R179C/H] in endocrine tumors or Gαs [R201C/Q] in hepatocellular carcinoma and pituitary tumors, and the latter by Gαq [Q209P] in uveal melanoma (cBioPortal for Cancer Genomics). These activating mutations have historically dominated the field of disease-related heterotrimeric G protein research. However, scatterings of point mutations with a dominant effect have emerged through whole-genome/exome sequencing throughout the amino acid sequences of Gα-subunits in diverse diseases, from pediatric encephalopathies to developmental abnormalities and immune system dysfunctions (Fig. 3). Interestingly, these pathogenic mutations are much less frequent in the AHD of the Gα-subunits. In contrast, non-pathogenic amino acid substitutions (polymorphisms) are found in the AHD of Gαo to a higher degree than in other regions of the protein (Fig. 2B) (Sun et al, 2024). We provide below some key and recent examples, using as reference the encephalopathic mutations found in Gαo, that is slowly becoming, from the pathogenic angle, one of the most studied Gα-subunits.

### GNAO1

The *GNAO1* gene encodes Gαo, a member of the inhibitory Gαi/o subclass and the major neuronal Gα-subunit across animal species (Sternweis and Robishaw, 1984). First described 12 years ago, dominant heterozygous de novo mutations in Gαo were associated with severe early-onset epileptic encephalopathy and Ohtahara

[1]Translational Research Center in Oncohaematology, Department of Cell Physiology and Metabolism, Faculty of Medicine, University of Geneva, Geneva CH-1211, Switzerland. [2]Translational Oncology Research Center, Qatar Biomedical Research Institute (QBRI), College of Health and Life Sciences, Hamad Bin Khalifa University (HBKU), Qatar Foundation, Doha PO Box 34110, Qatar. ✉E-mail: Vladimir.Katanaev@unige.ch; Gonzalo.Solis@unige.ch
https://doi.org/10.1038/s44321-025-00274-8 | Published online: 31 July 2025

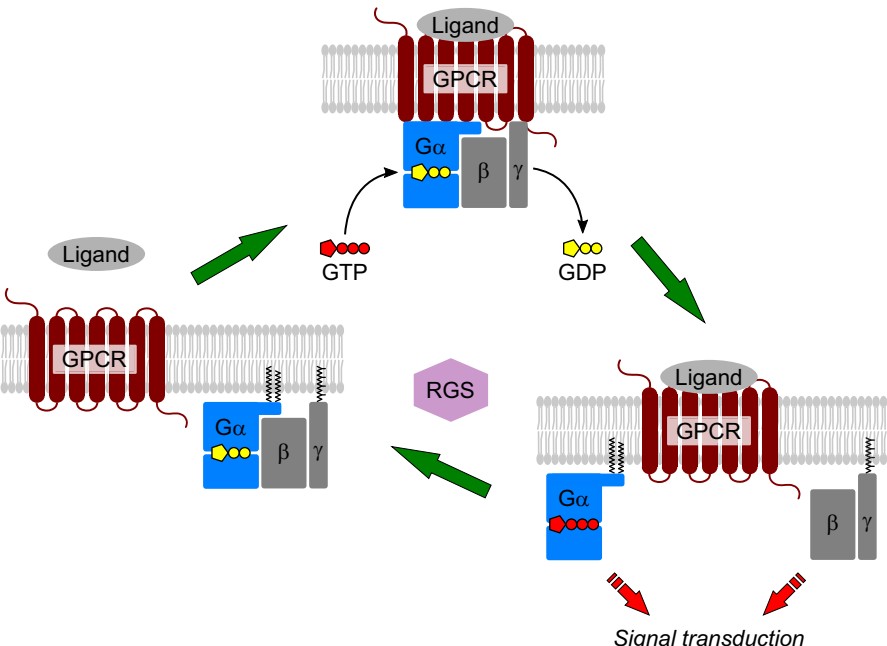

**Fig. 1. Heterotrimeric G protein cycle.**

Ligand-activated GPCR acts as a GEF, catalyzing the exchange of GDP for GTP on the Gα-subunit of the heterotrimeric G protein. Resultingly, the heterotrimer dissociates into Gα-GTP and Gβγ, both competent to transmit the signal further downstream. RGS proteins speed up GTP hydrolysis on Gα, switching the signaling off. Gα-GDP can re-associate with Gβγ.

syndrome (Nakamura et al, 2013). *GNAO1* encephalopathy typically manifests itself shortly after birth, and the patients present severe symptoms including profound intellectual disability, developmental delay, movement disorders, and intractable seizures. Subsequently, additional patients emerged presenting a movement disorder-predominant phenotype linked to a subset of recurrent *GNAO1* mutations (Menke et al, 2016). Thus, *GNAO1*-related pathologies were promptly classified as two distinct disorders in OMIM: "Developmental and Epileptic Encephalopathy 17" (DEE17; OMIM #615473) and "Neurodevelopmental Disorder with Involuntary Movements" (NEDIM; OMIM #617493). More recent pathogenic *GNAO1* variants leading to loss-of-function (LOF) and haploinsufficiency have been associated with milder phenotypes such as adolescent/adult-onset non-progressive dystonia, parkinsonism, and autism (Koval et al, 2023; Krenn et al, 2022; Lasa-Aranzasti et al, 2024a; Solis et al, 2025; Solis et al, 2024b; Wirth et al, 2022). Hence, it has been recently proposed that *GNAO1*-related disorders should be classified as a continuous phenotypic spectrum instead of two distinct phenotypes (Thiel et al, 2023). The distribution of some

DEE17 and NEDIM mutations over the structure of Gαo is provided in Fig. 2C; exclusion of the clinically severe mutations from the AHD and αN domains, and their clustering in the Ras-like domain, can be observed.

To date, multiple additional studies (Larasati et al, 2025b; Larasati et al, 2023; Lasa-Aranzasti et al, 2024a; Sáez González et al, 2023; Solis et al, 2021; Thiel et al, 2023) have identified >80 point mutations in >50 codons that lead to this dominant disease (Fig. 3). Most patients carry missense mutations, and only a few cases of deletions, frameshifts, duplications and short indels are known. Three different splice mutations in intron 6 have been described, of which c.724-8G>A leads to the in-frame insertion of two amino acids at the end of the Switch III region (Gαo [T241_N242insPQ], (Koval et al, 2023), while c.723+1G>A and c.723+2T>A—to an in-frame deletion of eight amino acids within the Switch III region (Gαo [V234_T241del], (Savitsky et al, 2025). Interestingly, the ClinVar archive contains >200 distinct entries just for missense *GNAO1* mutations, suggesting that *GNAO1*-related disorders are widely underreported.

While the amino acids highlighted in red in Fig. 3 reflect the position that could be

mutated in *GNAO1*-related disorders, dominant variants in this gene can also be found in other diseases/syndromes: [T327R] in severe pediatric speech deficiency (Hildebrand et al, 2020), [R243H] in triple-negative breast cancer (Kan et al, 2010), or [N312S] in diphtheria, tetanus and pertussis vaccination-associated seizures/epilepsy (Negi et al, 2023). Further, the same amino acids mutated in *GNAO1* encephalopathy can be mutated to cause other conditions. For instance, [R209C] has been characterized as a second-hit mutation leading to acute lymphoblastic leukemia in patients carrying the ETV6-RUNX1 gene fusion as the primary cancer predisposition (Song et al, 2021), while the [A227T] mutation is found in uterine endometrioid carcinoma and colon adenocarcinoma (cBioPortal for Cancer Genomics).

## GNAI1

*GNAI1* encodes Gαi1, a close relative of Gαo that also shows a prominent central nervous system expression. In a further similarity to Gαo, de novo point mutations in *GNAI1* have recently been described to cause dominant infantile neurological disorders jointly referred to as "Neurodevelopmental Disorder with Hypotonia, Impaired Speech, and Behavioral

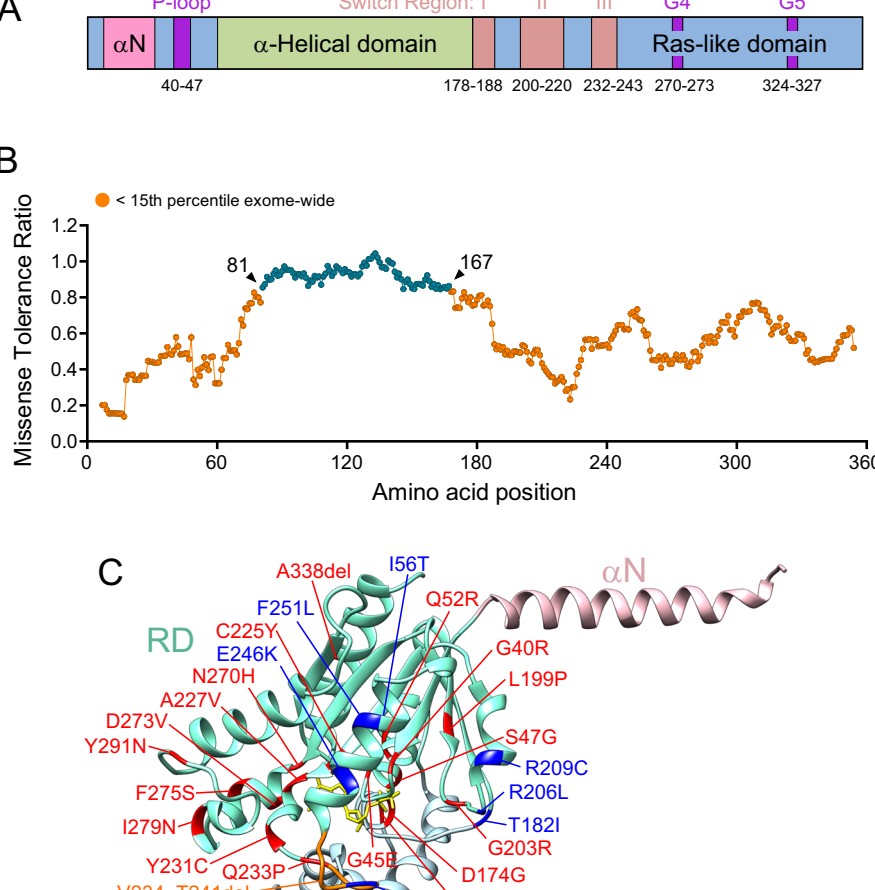

**Fig. 2. Gα-subunits' schematic structure and tolerance to non-pathogenic missense mutations.**

(A) Schematic organization of a Gα-subunit showing the three domains (αN, α-helical, and Ras-like domains) and motifs involved in guanine nucleotide binding and hydrolysis (P-loop, Switch I-III regions, G4 and G5) according to (Sprang, 1997). Amino acid boundaries are given on the example of Gαo. (B) Missense tolerance ratio (MTR, defined as the ratio of the observed to the expected proportion of missense variants adjusted by synonymous variation) of Gαo shows that the α-helical domain is most tolerant to missense variants in the genomes of ca. 1 million individuals. Variants with MTR values in the top-15-percentile exome-wide threshold (MTR < 0.841) have a higher probability to be deleterious. Data from https://rgc-research.regeneron.com (Sun et al, 2024). (C) Example of dominant mutations in Gαo leading to DEE17 (red) and NEDIM (blue) (Larasati et al, 2022; Larasati et al, 2025b; Larasati et al, 2023; Lasa-Aranzasti et al, 2024a; Savitsky et al, 2025; Solis et al, 2025; Solis et al, 2024a; Solis et al, 2021). Orange: DEE17-causing deletion of 8 amino acids. The homology model structure of Gαo was generated by the SWISS-MODEL server using as a template the crystallographic structure of Gαi1β1γ2 (1gp2) available at RCSB (rcsb.org). The N-terminal α-helix (αN), Ras-like domain (RD) and α-helical domain (AHD) are shown.

Abnormalities" (NEDHISB; OMIM #619854) (Muir et al, 2021). Overall, 12 codons were so far found mutated in *GNAI1*, 8 of which are the same as those found mutated in *GNAO1*-related disorders (Fig. 3). These mutations include [G40R], [G45D], [T48I/K], [Q52P], [D173V] (corresponding to [D174G] in *GNAO1*), and [D272G] ([D273V/Y] in *GNAO1*). Further, the [C224Y] mutation in *GNAI1* is mirrored by two novel pathogenic Gαo variants [C225Y/R] (Larasati et al, 2025b). The *GNAI1* variant

[P169R]—[P170R] in *GNAO1* (Larasati et al, 2023)—has been described in Prader-Willi-like syndrome with hypothyroidism and multiple pituitary hormone deficiency adding on top of the standard neurological manifestations (AlAli et al, 2024).

## GNAI2

Gαi2, the Gα-subunit encoded by the *GNAI2*, is another member of the Gαi/o

subfamily and displays a broad expression profile. Recently, several de novo mutations in *GNAI2* have been associated with massive developmental defects and immunological abnormalities leading to immunodeficiency and autoimmunity due to defective immune cell activation and migration (Ham et al, 2024). Remarkably, a significant overlap in the amino acids mutated in these dominant conditions can be found with the sites mutated in *GNAO1* and *GNAI1*

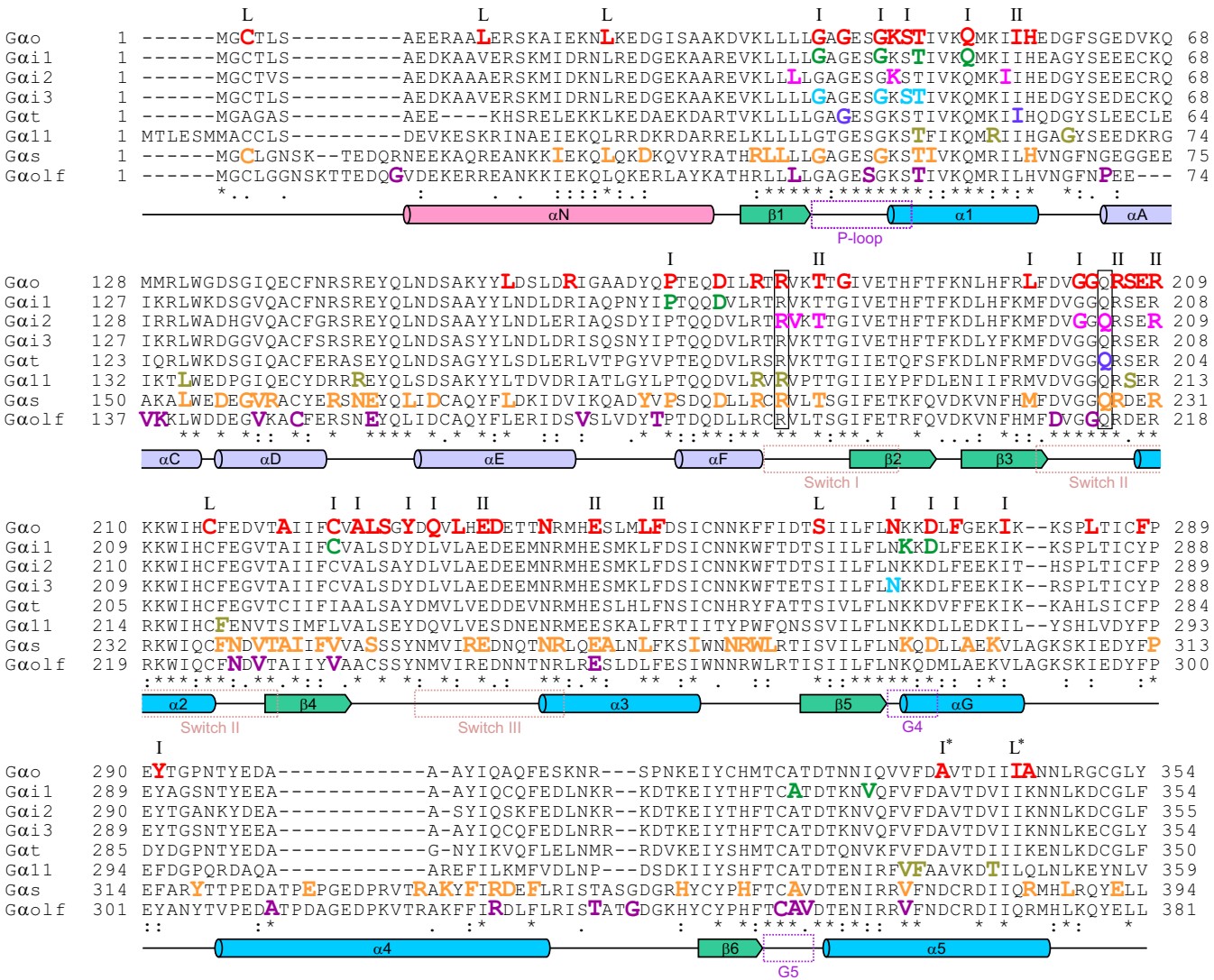

**Fig. 3. Alignment of the amino acid sequences of Gα-subunits with disease-causing dominant mutations (color-coded) was generated using the Clustal Omega server.**

Large parts (aa 69 to 127 for Gαo) of the α-helical domain are not shown, as this region rarely contains pathogenic mutations (see Fig. 2B,C). Mutations in the boxed Arg and Gln residues (R179 and Q205 in Gαo) lead to the GTPase-deficient pro-oncogenic forms of the G proteins. Conserved α-helices and β-sheets, Switch regions, and nucleotide binding P-loop, G4 and G5 domains are indicated according to (Sprang, 1997). Symbols above amino acids found mutated in *GNAO1*-related disorders highlight the sites that lead to non-neomorphic disease (L) and the neomorphic disease characterized by dominant interaction with Ric8A only (II) or both Ric8A and Ric8B (I) (* indicates a deletion mutation of the amino acid). Data from (Larasati et al, 2022; Larasati et al, 2023; Lasa-Aranzasti et al, 2024a; Solis et al, 2025; Solis et al, 2024a; Solis et al, 2021).

encephalopathies. Overall, nine codons in *GNAI2* have been identified to undergo missense mutations (Fig. 3), five of which have been equivalently reported mutated in *GNAO1*-related disorders: [K46T], [R179H/C], [T182A/I/P], [G203R], and [R209W].

### GNAI3

Dominant missense mutations in Gαi3, the member of the Gαi/o subclass encoded by *GNAI3*, have been associated with

"Auriculocondylar Syndrome-1" (ARCND1; OMIM #602483). ARCND1 is an autosomal dominant disorder of the first and second pharyngeal arches characterized by malformed ears (question mark ears), prominent cheeks, microstomia, abnormal temporomandibular joint, and mandibular condyle hypoplasia (Masotti et al, 2008). Notably, five codons in *GNAI3* have been linked to ARCND1 thus far (Fig. 3), all echoed in *GNAO1*-related disorders: [G40R], [G45A/V/S], [S47R/N], [T48N],

and [N269K/Y] ([N270D/H/K] in *GNAO1*) (Vegas et al, 2022).

### GNAT1

The missense mutations [G38D] ([G42R] in *GNAO1*), [I52N] ([I56T] in *GNAO1*), as well as the variant [K273del] ([K278del] in *GNAO1*) in *GNAT1* encoding transducin (Gαt), a retina-specific member of the Gαi/o subfamily, lead to "Congenital Stationary Night Blindness Autosomal Dominant-3"

(CSNBAD3; OMIM #610444) (Dryja et al, 1996). CSNBAD3 patients have normal daytime vision but are blind when light is so dim that requires rod-photoreceptors for vision. Alternative missense mutations in *GNAT1* have been associated with early- and late-onset high myopia (Zhou et al, 2018): [T73M], [R138H], [V170M], [I218T] — residues not described in *GNAO1*-related disorders—and [K273N], amino acid that has been found deleted ([K278del]) in *GNAO1* patients (Sáez González et al, 2023).

## GNAS

The *GNAS* gene encodes Gαs, the main Gα-subunit of the stimulatory Gαs subfamily and one of the most well-characterized G proteins. Canonical GTPase-deficient [R201C/Q] mutations in Gαs are recurrently found in hepatocellular carcinoma, lung adenocarcinoma, adrenocortical carcinoma, esophageal squamous cell carcinoma, and other tumors (cBioPortal for Cancer Genomics). The same mutations also underlie the McCune-Albright syndrome, which is characterized by endocrine disorders, polyostotic fibrous dysplasia, and café-au-lait skin spots. Similarly, the canonical GTPase-deficient Gαs mutants [Q227R/H] are found in endocrine tumors (Aldred and Trembath, 2000).

However, a long list of alternative dominant missense mutations in *GNAS* has been identified in a rare syndrome: Albright's hereditary osteodystrophy (AHO), characterized by skeletal and developmental abnormalities including brachydactyly, short stature, obesity, and mental deficits, with an accompanying parathyroid hormone resistance (pseudohypoparathyroidism type Ia (PHP1A; OMIM #103580) and Ic (PHP1C; OMIM #612462)) or without it (pseudopseudohypoparathyroidism (PPHP; OMIM #612463)), determined by the maternal *vs.* paternal inheritance of the pathogenic variant, respectively (Lemos and Thakker, 2015). Overall, >80 codons in *GNAS* have been found mutated in AHO, many of which correspond to variants in *GNAO1*-related disorders (Fig. 3): [L30P], [G47D], [G52D], [T55A], [H64L], [L179P], [P192L], [D196N], [R199G], [R201H/C/S/G/L], [T204I], [R228C], [R231C/H], [A243V], [E259V], [N264H], [E268K/G], [L272F], and [D295E] (Aldred and Trembath, 2000; Lemos and Thakker, 2015).

## GNAL

*GNAL* encodes Gαolf, the second member of the stimulatory Gαs subclass. Heterozygous mutations in *GNAL* lead to the autosomal dominant torsion Dystonia (Dystonia-25; OMIM #615073) with an onset in adulthood. Overall, 29 codons in *GNAL* are affected by missense mutations (Fuchs et al, 2013; Salamon et al, 2023), three of which correspond to positions found mutated in *GNAO1* patients (Fig. 3): [T57I] (Gαo [T48I]), [G213S] (Gαo [G204R/D]), and [E255A] that corresponds to the recurrent *GNAO1* mutations [E246K/G/R/V].

## GNA11

Missense mutations in *GNA11*—the gene encoding Gα11, a member of the Gαq subclass—have been associated with the autosomal dominant "Hypocalcemia-2" (HYPOC2; OMIM #615361) and "Hypocalciuric Hypercalcemia type II" (HHC2; OMIM #145981), both disorders of mineral homeostasis (Howles et al, 2023; Mannstadt et al, 2013). Of these mutations, several have their counterparts in *GNAO1*-related disorders (Fig. 3): [T54M] (Gαo [T48I]), [R181Q] (Gαo [R177P]), [R183P] (Gαo [R179G]), and [S211W] (Gαo [S207Y]). Further, the *GNA11* [A231T] mutation, equivalent to [A227V] in *GNAO1* encephalopathy (Fig. 3), could be found in glioblastoma multiforme, prostate adenocarcinoma, and other cancers (cBioPortal for Cancer Genomics).

This impressive and ever-expanding list of dominant missense mutations in *GNA* genes shows a systematic pattern of emergence across Gα-subunits (Fig. 3). This repeatability brings us to predict that many new pathogenic mutations will be discovered in *GNA*-dependent diseases, falling on the amino acid sites equivalent to the mutations already known in e.g., *GNAO1* encephalopathy (Fig. 3). The extent of such discoveries will be particularly important for the diseases, for which so far only a few missense *GNA* mutations have been identified, such as autosomal dominant congenital stationary night blindness (OMIM 610444: *GNAT1*), auriculocondylar syndrome-1 (OMIM 602483: *GNAI3*), autosomal dominant hypocalcemia-2 (OMIM 615361: *GNA11*), or dystonia-25 (OMIM 615073: *GNAL*). Furthermore, we predict that new congenital disorders are waiting to be discovered, which will be caused by dominant mutations in *GNA* genes so far not found in diseases: *GNAQ, GNAZ, GNA12, GNA13*, and others. The organs/tissues to be affected by these new congenital disorders to emerge are those where the expression levels of the respective Gα-subunits are high. Resources such as proteinatlas.org/ provide exhaustive information on expression patterns and abundances. Based on this, we may expect new *GNA12*-dependent deficiencies to be discovered that will affect the placenta, *GNA13*-dependent affecting lymphoid tissues, and *GNAZ*-dependent and *GNAQ*-dependent involving the cerebral cortex.

Other predictions emerging from the mutational signatures found in pathogenic missense *GNA* mutations relate to unifying mechanisms of pathogenic dominance driven by these mutations in different Gα-subunits, and to pharmacological avenues to treat the diverse Gα-pathies, as elaborated in the following sections.

## Guanine nucleotide mishandling as the unifying biochemical deficiency of dominant *GNA* mutations

As a G protein, the Gα-subunit binds the guanine nucleotides GDP and GTP. It adopts an activated conformation able to interact with and activate downstream signaling partners when bound to GTP, and it hydrolyzes GTP to GDP to become inactive again (Fig. 1). These activities are the biochemical basis of the Gα life cycle. A group of proteins further regulates these activities: Gβγ and GoLoco domain-containing proteins act as GDIs (guanine nucleotide dissociation inhibitors), activated GPCRs and certain non-receptor proteins act as GEFs, and dedicated proteins, most notably RGS proteins, act as GAPs.

Some of the first attempts to elucidate the biochemical consequences of pathogenic *GNAO1* mutations were performed using recombinant Gαo for the recurrent variants [R203R], [R209C/H], and [E246K] (Larasati et al, 2022; Larrivee et al, 2020). This approach revealed a major disruption in the GDP/GTP cycle, showing two emerging defects: (i) a much faster GTP uptake (which reflects faster GDP release), and (ii) a strongly reduced GTP hydrolysis. This evidence suggests that, biochemically, *GNAO1* mutations lead to the constitutively GTP-binding state of Gαo. A similar pattern was later reproduced in a large number of Gαo mutants (Knight et al, 2023; Lasa-Aranzasti et al, 2024a; Solis et al, 2024a). A strong perturbation in the GDP/GTP cycle appears to be exclusively induced by *GNAO1* mutations associated with DEE17 and NEDIM, while Gαo mutants linked to

milder phenotypes presented a near-normal GDP/GTP cycle (Knight et al, 2023; Koval et al, 2023; Solis et al, 2025; Solis et al, 2024b). Further, several Gαo variants leading to the most severe DEE17 phenotype were completely incapable of binding GTP (and perhaps any nucleotide) (Knight et al, 2023; Larasati et al, 2025b; Larasati et al, 2023; Solis et al, 2024a; Solis et al, 2021).

Interestingly, some pathogenic mutations in other members of the Gαi/o subfamily showed similar biochemical defects. For instance, almost the entire set of Gαi2 mutants leading to impaired immunity showed a faster GTP-uptake and decreased hydrolysis (Ham et al, 2024). The Gαi1 [Q52P] and Gαi3 [S47R] variants associated with NEDHISB and ARCND1, respectively, were unable to bind to GTP (Marivin et al, 2016; Solis et al, 2021). The remaining ARCND1 mutants of Gαi3 were not properly expressed in bacteria (Marivin et al, 2016); other pathological Gαi1 variants have not been analyzed biochemically. On the other hand, the Gαt [G38D] mutant responsible for the Nougaret form of night blindness showed a near-normal GTP-binding, and a slight reduction in both intrinsic and RGS9-stimulated GTPase activity (Muradov and Artemyev, 2000), although this study employed a Gαt/Gαi1 chimera comprised of 94% of Gαt residues.

While Gα11 and Gαolf mutants have not been biochemically characterized yet, several studies describing defects in the GDP/GTP cycle of pathogenic Gαs variants exist. Specifically, an accelerated GTP-uptake was described for Gαs [R228C], [R265H], [W281R], [A366S], and [A366_T369insAVDT] (Hu and Shokat, 2018; Iiri et al, 1994; Jeong and Chung, 2023; Makita et al, 2007), although several other pathogenic Gαs variants display normal GTP binding (Jeong and Chung, 2023). An abolished GTP-loading was reported for [I106S], [D173N], [R231H], and [R280G] (Iiri et al, 1997; Jeong and Chung, 2023; Leyme et al, 2014). Several other Gαs variants—[L99P], [F246S], [S250R], [R258W], [E259V], and [K338N]—were expressed in aggregates in *E. coli* (Jeong and Chung, 2023), suggesting defects in the proper folding of the protein. In fact, three of these variants ([S250R], [R258W], and [E259V]) were previously described as thermolabile, as they aggregate at physiological temperature (Warner et al, 1997; Warner et al, 1999; Warner et al, 1998).

If nucleotide handling is strongly affected in pathogenic Gα variants, their cellular interactions, that depend upon the well-adopted and distinct for each bound nucleotide (GDP *vs.* GTP) states, are expected to be affected too. This is indeed the case, and we will now discuss these cellular deficiencies, grouping them into the interactions with Gβγ-heterodimers and GPCRs (both recognize the GDP-bound state of Gα) and with RGS proteins and effectors (that recognize the GTP-bound state).

## Functional consequences of dominant *GNA* mutations

Gβγ-heterodimers are the main binding partners of Gα-subunits (Pierce et al, 2002), and defects in the formation of the heterotrimeric G protein are present in most Gαo variants connected to *GNAO1*-related disorders. Particularly, Gαo mutants leading to DEE17 showed a very weak association with Gβγ, which strongly correlates with their poor localization at the plasma membrane (Domínguez-Carral et al, 2023; Larasati et al, 2023; Lasa-Aranzasti et al, 2024a; Solis et al, 2024a; Solis et al, 2021). Gαo variants related to NEDIM and milder phenotypes tend to interact with Gβγ to a near-normal level or even showed a stronger binding. It was thus postulated that the degree of impairment in heterotrimer formation serves as a predictor of clinical severity in *GNAO1* mutations (Domínguez-Carral et al, 2023; Solis et al, 2024a). Gβγ-binding, however, does not fully account for disease severity, as mutations targeting the αN domain of Gαo lead to a mild Parkinsonism phenotype despite failing to form the heterotrimer (Solis et al, 2024b). Moreover, several DEE17 and NEDIM variants appear to sequester Gβγ due to a failure in adopting the GTP-induced conformational changes needed for their dissociation upon GPCR activation (Knight et al, 2023). Additional Gαo variants associated with mild phenotypes are also predicted to sequester Gβγ as they exhibit a poor coupling with GPCRs despite a near-normal formation of the heterotrimer (Lasa-Aranzasti et al, 2024a; Solis et al, 2025).

Albeit its relevance, only a few studies have directly addressed the interaction with Gβγ for other pathogenic Gα-subunits. For instance, mild to moderate defects in Gβγ-binding were reported for all Gαi3 mutants linked to ARCND1 (excepting [S47R] with normal binding (Marivin et al, 2016)), and

for some Gαolf variants leading to Dytonia-25: [F133L], [V137M], [E155K], [G213S], and [A353T] (Dos Santos et al, 2016; Fuchs et al, 2013; Kumar et al, 2014). The autosomal recessive Gαolf [R329W] also displayed a mild reduction in Gβγ interaction (Masuho et al, 2016). Lack of in vitro association with Gβγ was shown for Gαs [S250R] and [E259V] (Warner et al, 1997; Warner et al, 1999), whereas no binding defect was described for [R231H] (Iiri et al, 1997). Finally, the Nougaret Gαt [G38D] variant also showed a normal interaction with Gβγ (Muradov and Artemyev, 2000).

Gαo variants were first reported as either normal, LOF, or gain-of-function (GOF) according to their ability to inhibit the forskolin (FSK)-mediated production of cAMP downstream of the $\alpha_{2A}$-adrenoceptor ($\alpha_{2A}$AR) (Feng et al, 2017). Conversely, later analyses employing various biosensors to monitor GPCR-mediated signaling via the heterotrimeric Go protein—i.e. the splitting of Gαo and Gβγ—showed that the clinically severe *GNAO1* mutations were impaired to various degrees in their activation by GPCRs, while milder mutations showed minor to negligible effects (Domínguez-Carral et al, 2023; Knight et al, 2023; Koval et al, 2023; Muntean et al, 2021). The extent of these defects, however, depends on the exact GPCR as the recurrent Gαo variants [G203R], [R209C] and [E246K] showed dissimilar LOF behaviors upon stimulation by the $\alpha_{2A}$AR, dopamine $D_2$ ($D_2$R), μ-opioid (MOR), and $M_2$ muscarinic acetylcholine ($M_2$R) receptors (Larasati et al, 2022). Additional *GNAO1* mutations associated with mild phenotypes lead to LOF by the exclusion of the mutant Gαo from GPCR-signaling, due to the lack of heterotrimer formation (Solis et al, 2024b), deficiency in coupling with GPCRs (Lasa-Aranzasti et al, 2024a), or poor targeting to the plasma membrane (Lasa-Aranzasti et al, 2024b). Similar studies for *GNAI3* mutations revealed that all tested pathogenic Gαi3 variants were poorly activated by the adenosine $A_1$ receptor ($A_1$R) (Marivin et al, 2016). Severe deficiencies in G protein activation were also reported for Gαs [E268K] and Gαolf [F133L], [G213S], and [A353T] downstream of the $\beta_2$-adrenoceptor ($\beta_2$AR) and dopamine $D_1$ receptor ($D_1$R), respectively, whereas Gαolf [E155K] showed mild defects and Gαolf [V137M] was normally activated (Dos Santos et al, 2016; Fuchs et al, 2013; Knight et al, 2021; Kumar et al, 2014).

Downstream functional analyses have also been performed for pathogenic Gα-subunits. For instance, the sole expression of Gαi2 mutants significantly reduced FSK-mediated cAMP production in HEK293 cells, suggesting a GOF towards adenylate cyclase inhibition. Although this effect was not seen for Gαi2 [R209W] and [G203R], FSK-mediated cAMP production was decreased in patients' primary fibroblasts carrying the [R209W] variant. When chemokines were used to counterbalance FSK-mediated cAMP production, only a few Gαi2 variants slightly reduced cAMP levels, suggesting that the remaining mutants cannot engage with GPCRs. A strong reduction in coupling to the CC-chemokine receptor 7 (CCR7) was indeed confirmed for Gαi2 [T182A] using a bioluminescence resonance energy transfer (BRET)-based assay (Ham et al, 2024). In contrast, the active GTP-loaded form of Gαt [G38D] lost the ability to bind and activate cGMP-specific phosphodiesterase 6, pointing to a LOF of this variant downstream of GPCR stimulation (Moussaif et al, 2006; Muradov and Artemyev, 2000). Various extracellular calcium-sensing assays have been employed to characterize pathogenic Gα11 mutants, classifying the mutants [L135Q], [I200del] and [T347A] associated with HHC2 as LOF and the HYPOC2-variants [R60C/L], [R181Q], [S211W], and [F341L] as GOF (Boisen et al, 2025; Mannstadt et al, 2013; Nesbit et al, 2013).

Despite some contrasting biochemical properties, all pathogenic Gαs variants lead to a LOF phenotype through distinct molecular mechanisms. Specifically, a subset of Gαs mutants—[R258W], [F376V], [R385H], [L388R], [Y391X], [E392K], and [E392X]—are inefficient or totally excluded from activation via the β2AR and/or parathyroid hormone type 1 receptor (PTH1R), although they retain the ability to stimulate adenylate cyclases (Biebermann et al, 2019; Schwindinger et al, 1994; Seven Menevse et al, 2024; Thiele et al, 2011; Warner et al, 1998). Gαs [I382del] was first reported as partial LOF for PTH1R but not for other receptors, including β2AR (Wu et al, 2001), yet later it was shown to be insensitive for both GPCRs (Linglart et al, 2006). For another subset of variants—[R228C], [R231H], [S250R], [E259V], [R265H], [K338N], and [L388P]—the LOF phenotype was due to their failure to activate adenylate cyclases (Cleator et al, 2004; Farfel et al, 1996; Hu and Shokat, 2018; Iiri et al, 1997;

Leyme et al, 2014; Pohlenz et al, 2003; Thiele et al, 2011; Warner et al, 1997; Warner et al, 1999). A few additional Gαs variants—[L99P], [R165C], [H362P], [A366S], and [A366_T369insAVDT]—were described as thermolabile and/or poorly expressed (Iiri et al, 1994; Linglart et al, 2006; Makita et al, 2007; Miric et al, 1993).

Probably one of the most striking aspects of several GNAO1 mutations is that they lead to a dominant GPCR-coupling. Using a BRET-based assay that directly monitors GPCR engagement by the Gα-subunit, we showed that many of the clinically severe Gαo mutants coupled normally or even strongly with $M_2R$ despite a poor Gβγ-binding (Solis et al, 2024a). Many of these variants were also shown to block the activation of the wild-type Gαo in a different BRET-based assay that measures Gβγ-disengagement upon $D_2R$ stimulation (Domínguez-Carral et al, 2023; Muntean et al, 2021). A dominant GPCR-coupling was recently confirmed for Gαo [K46E] that forms an inactive stable complex with $D_2R$ and Gβγ as revealed by cryo-electron microscopy (Knight et al, 2024), and appears to be a phenomenon general for severe variants (Larasati et al, 2025a). Interestingly, all five ARCND1-associated Gαi3 mutants were shown to block the endothelin type A receptor ($ET_AR$), excluding Gαq from signaling mediated by endothelin-1 (Marivin et al, 2016). This suggests that a dominant GPCR-coupling might be prevalent among Gα-pathies, as these Gαo and Gαi3 mutations affect highly conserved residues often mutated in other Gα-subunits (Fig. 3). Moreover, several experimental, not derived from patients, mutations in Gαi1, Gαi2, Gαs and Gαt have been shown to engage GPCRs in a similar dominant fashion (Kaya et al, 2016; Liang et al, 2018). The dominant GPCR-coupling of pathogenic Gαo variants, however, does not correlate with the severity of the disease (Domínguez-Carral et al, 2023; Solis et al, 2024a), implying that additional factors are at play.

The examples above illustrate that different pathogenic Gα variants have varying extents of deficiencies in the interactions with Gβγ, GPCRs, and effectors, likely reflecting their defects in nucleotide handling and adoption of proper conformations. Perhaps the most telling, in this regard, is the deficiency of the pathogenic variants' interactions with RGS proteins – the regulatory proteins that recognize the well-

adopted activated, GTP-bound state of Gα (Ross and Wilkie, 2000). Excluding only a few variants associated with mild phenotypes (Lasa-Aranzasti et al, 2024a; Solis et al, 2025; Solis et al, 2024b), Gαo mutants showed a drastically reduced interaction with RGS19, a major regulator of GTP hydrolysis on Gαo (Larasati et al, 2022; Larasati et al, 2023; Lin et al, 2014; Solis et al, 2024a; Solis et al, 2021). Similarly, pathogenic Gαo interactions with RGS4 (Koval et al, 2023), pathogenic Gαi2 with RGS16 (Ham et al, 2024), and pathogenic Gαt with RGS9 (Muradov and Artemyev, 2000) were defective. Thus, despite the biochemically constitutive GTP loading by many pathogenic variants, they fail to adopt the proper activated conformation (Larasati et al, 2022; Solis et al, 2024a).

## The Neomorphic nature of the dominant GNA mutations

When assessing the dominant phenotypes emanating from the pathogenic GNAO1 variants, we were puzzled that the same variant's activities, in different systems, could be (and were by different authors) interpreted as LOF, dominant-negative (DN), or GOF. For example, the Gαo [G203R], [E246K], and [R209C] variants have reduced (to different extents) capacity to transduce signals from neuronal GPCRs, arguing for a partial LOF nature of these mutants. Similarly, in the Drosophila model of GNAO1 encephalopathy, $Gαo^{G203R}/Gαo^{G203R}$ homozygous mutant flies die at the second larval stage, later than the $Gαo^{-/-}$ null animals that do not survive past the embryonic stage, also suggesting the partial LOF genetics of the mutation (Larasati et al, 2022). However, the same mutation can be classified as GOF, due to its massively increased rate of GTP uptake accompanied by the loss of GTP hydrolysis (Larasati et al, 2022) or its higher ability to inhibit adenylyl cyclase (Feng et al, 2017). Finally, the same mutation behaves in other cellular assays, and also in diverse animal models, as a DN (Di Rocco et al, 2021; Larasati et al, 2022; Muntean et al, 2021; Silachev et al, 2022; Wang et al, 2022). Based on all the contrasting properties of pathogenic Gαo, we proposed that GNAO1 mutations are neomorphic in nature (Larasati et al, 2022) as originally defined by the Nobel laureate Hermann Muller: "*neomorph represents a change in the nature of the gene… giving an effect not produced, or at*

*least not produced to an appreciable extent, by the original normal gene"* (Muller, 1932).

While searching for the neomorphic interactor/transducer of pathogenic Gαo, we took into consideration the following observations: (i) several Gαo mutants are biochemically inactive at room temperature after recombinant production (Solis et al, 2024a; Solis et al, 2021); (ii) most mutants have decreased cellular expression levels (Domínguez-Carral et al, 2023; Feng et al, 2017; Knight et al, 2023; Larasati et al, 2025b; Solis et al, 2024a); (iii) despite constitutive GTP loading, the mutants uniformly fail to interact with RGS proteins that normally recognizes the GTP-bound state of Gα-subunits (Ham et al, 2024; Koval et al, 2023; Larasati et al, 2022; Larasati et al, 2023; Lasa-Aranzasti et al, 2024a; Solis et al, 2021). All these features hint at a potential folding problem, which prompted us to investigate Ric8 proteins.

Initially described as cytosolic GEFs, Ric8A/B have also emerged as mandatory chaperones for Gα-subunits (Gabay et al, 2011), with Ric8A responsible for the Gαi/o, Gαq, and Gα12/13 subfamilies and Ric8B exclusively for Gαs/olf. We discovered that clinically severe *GNAO1* variants massively sequester Ric8A and relocalize it to the Golgi apparatus. Remarkably, the same was also observed for Ric8B, the chaperone "foreign" for the Gαi/o subfamily. Furthermore, the strength of the neomorphic binding to and Golgi-relocalization of Ric8B correlates with the clinical severity of the encephalopathy, emerging as the best molecular biomarker of the disease (Katanaev and Valnohova, 2025; Solis et al, 2024a). Importantly, the neomorphic sequestration of Ric8A/B by pathogenic Gαo interferes with the Ric8 chaperone activity over several Gα-subunits, leading to a drop in their expression (Solis et al, 2024a). This revealed an unanticipated mechanism of disease dominance in *GNAO1*-related disorders: clinically severe variants, in addition to the DN action towards the wild-type copy of Gαo itself, might infringe the entire GPCR-signaling network in the brain, the tissue in which Gαo is most abundantly expressed. Based on the strength of the neomorphic Ric8 binding (Larasati et al, 2025b; Solis et al, 2025; Solis et al, 2024a; Solis et al, 2024b), *GNAO1* mutations can be broadly grouped in three main categories (Fig. 3). Variants presenting a strong gain of binding to both Ric8A and Ric8B define the first category (I) that includes mutations associated almost exclusively with the most severe DEE17 phenotype. The second category (II) involves NEDIM mutations that result in a strong binding to Ric8A but intermediate to Ric8B. And Gαo mutants carrying a near-normal Ric8 association define the third category (L) that contains partial LOF *GNAO1* mutations leading to mild phenotypes such as late-onset dystonia and Parkinsonism.

Remarkably, an increased binding to Ric8A was also detected for all five ARCND1-associated Gαi3 mutants in a yeast two-hybrid assay (Marivin et al, 2016), although the authors speculated that this was related to Ric8 GEF activity. Additionally, a failure in adopting the active conformation was demonstrated for the recombinant Gαi3 [S47R] due to its high susceptibility to trypsinolysis when loaded with a non-hydrolysable GTP analog (GTPγS). A similar lack in acquiring the active conformation upon GTPγS-loading and trypsinolysis was long established for some pathogenic Gαs variants: [D173N], [S250R], [E259V], and [R265H] (Leyme et al, 2014; Warner et al, 1997; Warner et al, 1999). Our ongoing analysis of selected pathogenic variants of Gαs and Gαi1 points to a gain of binding to Ric8A/B by the mutant Gα-subunits (unpublished). We thus postulate that a gain of neomorphic Ric8 binding due to folding defects might be at the core of many Gα-pathies—particularly, in their most severe manifestations. Thus, the mechanism of dominance seen for *GNAO1*-related disorders may underlie other Gα-pathies, with the main tissue/organ affected by the disease determined by where the given Gα-subunit is expressed to the highest levels.

The features detailed in these chapters point to our second major prediction: that clinically severe variants across diverse Gα-pathies—from neurological to metabolic, and from immunological to developmental disorders—share unifying molecular and cellular mechanisms. These include deficient guanine nucleotide handling, dominant GPCR-coupling, and neomorphic sequestration of the Ric8A/B chaperones. Together, these dysfunctions may exert a dominant effect on the whole GPCR-signaling network.

## Unified drug discovery for Gα-pathies

Gα-pathies are currently managed through treatments that aim to control clinical manifestations rather than correct the underlying genetic defect. For example, patients with *GNAO1* and *GNAI1* mutations are treated with anti-seizure medications to manage epilepsy and movement disorders (Muir et al, 2021; Sáez González et al, 2023), and dystonia is treated pharmacologically in *GNAL* patients (Salamon et al, 2023). *GNAI2*-related symptoms are managed with antibiotics, growth hormone and/or testosterone injections, depending on the clinical presentation (Ham et al, 2024). *GNAI3*-related malformations are addressed with cosmetic and corrective procedures (Vegas et al, 2022). *GNAS* mutations resulting in hormone resistance syndromes are managed with supplementation (e.g., calcium, vitamin D, magnesium), dietary guidance, and growth hormone therapy (Lemos and Thakker, 2015). *GNA11*-associated disorders are primarily treated by regulating calcium levels (Howles et al, 2023).

Notably, the development of CRISPR-Cas9 gene editing technology led to the first FDA-approved gene therapy for sickle cell disease and β-thalassemia (Yen et al, 2024). This breakthrough paves the way for treating other genetic disorders, with *GNAI2* mutations in T cells and *GNAT1* mutations in retinal photoreceptors emerging as promising candidates for targeted gene correction. Moreover, AAV-mediated delivery of RNA interference (RNAi) is currently being explored as a potential therapeutic strategy for allele-specific silencing in *GNAO1*-related disorders (Shomer et al, 2025).

Based on the idea of shared mechanisms across pathogenic Gα-subunits, our third major prediction states that a unifying drug discovery paradigm can be expected, whereas drugs discovered to be effective against one Gα-pathy may turn out to be effective also against the broad scope of Gα-diseases. As detailed in the previous sections, the following molecular aspects of pathogenic Gα variants can be identified as key to the molecular etiology of the diseases they cause: (1) defective guanine nucleotide handling (GTP uptake and hydrolysis); (2) neomorphic interactions with Ric8A/B; (3) dominant non-productive interactions with and sequestration of GPCRs.

Each of these features can be targeted with drug discovery. Having established a high-throughput screening (HTS) platform, we screened a library of FDA-approved drugs and identified zinc salts as the agent capable of rescuing the GTPase deficiency of pathogenic Gαo mutants without affecting Gαo wild-type (Larasati et al, 2022). Subsequent investigations uncovered the

molecular mechanism of Zn²⁺: substituting Mg²⁺ in the active center of pathogenic Gαo, zinc could bring back the Gln205 side chain, otherwise swept away from the interaction with the γ-phosphate of GTP, needed for the GTP hydrolysis reaction. Restoration of deficient cellular interactions, and partial rescue of the motor activities and life span in the *Drosophila* model of *GNAO1* encephalopathy was observed upon dietary zinc supplementation (Larasati et al, 2022). Demonstration of the safety of zinc supplementation in mouse models of the disease—and the approved nature of oral zinc treatment for a number of pediatric conditions, such as the Wilson disease—permitted to swiftly pass to the first-in-human application of the drug to a 3-year-old patient with severe encephalopathy, which resulted in strong improvements of patient's conditions (Larasati et al, 2025c). Currently, larger-scale clinical applications of zinc to treat *GNAO1* encephalopathy are in place in Germany (ClinicalTrials.gov ID: NCT06412653) and China. This is a rare success story, with the drug to pass from HTS-based discovery to clinical applications in ca. 2 years (Larasati et al, 2022; Larasati et al, 2025c). Importantly, we now find that zinc can similarly restore the defective GTPase activity in pathogenic Gαi2 variants (unpublished observations), laying the ground for development of the zinc supplementation therapy, first in animal models and then hopefully in clinical trials, providing hope for this novel devastating pediatric disease caused by dominant Gαi2 mutations (Ham et al, 2024). We expect that, similarly, zinc may prove effective in other Gα-pathies, which will provide a long-required treatment option for a wide diversity of these rare diseases.

We expect that other drugs may emerge that could act at the defective guanine nucleotide handling by pathogenic Gα variants. Similarly, drug discovery platforms can be designed to target the other two identified molecular vulnerabilities in Gα-pathies, i.e., neomorphic sequestration of Ric8A/B and the dominant engagement of GPCRs seen for some pathogenic variants. And again, the drug candidates to emerge from these discovery campaigns may prove to be effective across diverse Gα-pathies.

## Conclusions

By intention, in the sections above, we streamlined the hypothesis of the unifying mechanisms of Gα-pathies, from molecules to patients, in a simplified manner. Five unifying elements constitute these mechanisms. The first is a disruption of the guanine nucleotide handling capacity by pathogenic Gα variants. The second (as the basis of the first one) is an increased structural flexibility and decreased folding of the pathogenic Gα variants. The third (as the cellular consequence of the second one) is the neomorphic binding of pathogenic Gα-subunits to Ric8A/B proteins and the dominant coupling with GPCRs, resulting in the misconfiguration of the whole cellular Gα- and GPCR-signaling network. The fourth (as the consequence of the third one at the organismal level) is the manifestation of the disease in the specific organs and tissues with high expression levels of the given Gα. And the fifth element is that despite the diversity of the clinical manifestations, the same drugs can be applicable across Gα-pathies. However, the complexity of the 16 Gα-subunits, the numerous pathogenic mutations scattered around their sequences, and the diverse disease manifestations these mutations cause are larger than this chain of mechanisms. Thus, we consider these unifying mechanisms to be the "backbone" of the complex molecular-clinical picture of Gα-pathies. We further expect that the many details of (mys)functioning of the many pathogenic variants generate a multiparametric matrix, which populates the "meat" on top of this unifying backbone, to ultimately comprehend the complexity of the many rare diseases we overview here—and to provide the basis for the translational breakthroughs for the treatment of these diseases.

A deeper analysis of pathogenic Gα mutations reveals unifying mechanisms—the backbone of Gα-pathies—which may underlie rare and complex disorders. These patterns can help predict novel disease-causing variants in *GNA* genes not previously linked to any pathology. The shared features among these mutations also shed light on common mechanisms of pathogenic dominance and open possibilities for therapeutic strategies that could apply across different disease phenotypes. These insights have the potential to extend beyond heterotrimeric G proteins, offering broader relevance to the study of their homologs, such as the small Ras-like G proteins. Thus, dominant Gα mutations share common biochemical/cellular dysfunctions, common mechanisms of dominance, and common avenues of therapeutic correction in neurological, immune, metabolic, and developmental diseases, caused by mutations in the 16 genes encoding human Gα-subunits—the major transducers of the main eukaryotic receptor family.

## Peer review information

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

## Author contributions

**Vladimir L Katanaev**: Conceptualization; Formal analysis; Supervision; Investigation; Methodology; Writing—original draft; Project administration; Writing—review and editing. **Gonzalo P Solis**: Data curation; Formal analysis; Investigation; Visualization; Methodology; Writing—original draft; Writing—review and editing.

## Disclosure and competing interests statement

The authors declare no competing interests.

