## [Peer Review File · EMBO Molecular Medicine]

Dominant G α mutations in human disease: unifying mechanisms and treatment strategies

Vladimir Katanaev and Gonzalo Solis

Corresponding authors: Vladimir Katanaev (vladimir.katanaev@unige.ch) , Gonzalo Solis (Gonzalo.Solis@unige.ch)

Review Timeline:

Submission Date:	5th Feb 25
Editorial Decision:	3rd Mar 25
Editor's Correspondence:	19th May 25
Revision Received:	2nd Jun 25
Editorial Decision:	27th Jun 25
Revision Received:	28th Jun 25
Accepted:	4th Jul 25

Editor: Lise Roth

Transaction Report:

3rd Mar 2025

Dear Prof. Katanaev,

Thank you for the submission of your Perspective to EMBO Molecular Medicine. We have now received feedback from the experts who agreed to evaluate your manuscript.

As you will see from the reports below, while referee #1 is overall positive and has only minor suggestions, referee #2 is more critical and suggests a major rewriting of the manuscript, which would include stepping away from the idea of a predictive table or model.

After discussion within the team, we would welcome a revised version of your manuscript that would address the referees' points. Please attach a covering letter giving details of the way in which you have handled each of the points raised by the referees.

- 1/ A .doc formatted version of the manuscript text (including Figure legends and tables)
- 2/ Separate figure files.
- 3/ A letter INCLUDING the reviewer's reports and your detailed responses to their comments.
- 4/ A 'disclosure statement and competing interests' statement (<https://www.embopress.org/competing-interests>).
- 5/ Author contributions: CRediT has replaced the traditional author contributions section because it offers a systematic machine readable author contributions format that allows for more effective research assessment. Please remove the Authors Contributions from the manuscript and use the free text boxes beneath each contributing author's name in our system to add specific details on the author's contribution. More information is available in our guide to authors.
- 6/ The Summary should be labelled abstract.
- 7/ While we are flexible, we recommend a maximum of 50 references.

For the figures, please note the following points:

- If there are certain aspects of your figure draft that are based upon assumptions or where the scientific data remains ambiguous, please add a comment so that we can work with you on an accurate depiction. Please ensure the directionality and nature of interactions is presented accurately.
- If the figure or single panels of the figure have been adapted from a published figure, please add this information to the figure legend (e.g., 'Adapted from...' or 'Based on...').
- Please only re-use figures or parts of a figure if this is essential for understanding the concept communicated. Often a reference to a previous paper will suffice. If the figure contains re-used images or elements of images, please make sure that you have the permission/license to publish it. All re-used material must be explicitly cited.
- If you use an image data base for scientific iconography (e.g., BioRender), please let us know if you have a license that allows for publication in an academic journal.

Looking forward to receiving your revised manuscript,

With kind regards,

Lise Roth

***** Reviewer's comments *****

Referee #1 (Remarks for Author):

The review article by Katanaev and Solis discusses a very interesting topic. The periodic table of dominant Galpha mutations - is very exciting. It has a highly useful predictive power: it tells which mutations will be discovered in the future in the Galpha genes already known to cause disease, and even which new rare diseases will emerge in the future caused by mutations in the Galpha genes not previously associated to any pathogenic variants. This concept further predicts the unifying molecular mechanism of the diverse diseases caused by Galpha mutations. And finally, it predicts that the same drug can be proven applicable and useful across the spectrum of Galpha-pathies, despite these diverse diseases (developmental, oncogenic, metabolic, neurological, immune, etc.).

Expanding on the therapeutic side and including potential approaches other than small molecules would be helpful and informative. Although CRISPR and base editing are specific to some mutations, they could still be relevant to treating some of the diverse diseases. This addition and discussion would be highly recommended for publications in EMM.

Referee #2 (Remarks for Author):

The review by Katanaev and Solis summarizes the information about mutations across all Ga subtypes into a unified table of predictable mutations. The authors focus on mutation position, nucleotide exchange, hydrolysis, and gain/loss of function to argue mutations in disparate G alpha subunits have predictable biochemical and functional outcomes. The authors also focus on so-called neomorphic gain of function mutations which they themselves characterized in previous publications and define as increased binding to known chaperone proteins ric8a/b. Three main postulates are formed at the end of the article: 1) New mutations will be discovered at identical positions to those already characterized in GNAO1 encephalopathy; 2) Nucleotide mishandling is a unifying molecular mechanism for Ga pathologies 3) A unified drug discovery paradigm can be expected whereby drugs effective for one Ga type can be applied to many more.

The authors go to great lengths to characterize mutations from multiple Ga subtypes that have arisen over decades of research and attempt to explain the wide variety of clinical and cellular phenotypes with the above postulates.

A comprehensive picture of diverse pathological Ga missense mutations is a significant addition to clinicians and G-protein field specialists interested in these rare diseases. However, the article as presented has serious flaws.

1. The concept of periodic table is grossly oversold. At its core, for anyone studying G proteins the concept of conserved structural elements in Ga subunits doing conserved things and their disruptions having easily predictable effects consistent with their functional role in G proteins (e.g. interaction with GPCRs, nucleotide binding and hydrolysis, association with Gbg, etc) conserved across different Ga isoforms is foundational. It is firmly in place based on results of many labs over decades and at this point rather trivial that does not require "Mendeleev" to put a stake in it. If you go beyond this concept - one very quickly gets into the fast sand and things become complicated beyond an ability to present a clear "periodic table". The principles for organizing such a table were not made clear by the authors and they actually do not include any table, beyond surface level common knowledge arguments. Rather, the authors provide lists of overarching mechanisms, which themselves are poorly defined and inconsistent. Furthermore, competing language such as "unifying mechanism" and "multiparametric matrix" to describe the proposed periodic table highlight the paradoxical nature of making a model simultaneously reductive and nuanced. Therefore, is unclear whether the authors are ultimately arguing that Ga pathologies should be analyzed along several dimensions, or whether there is a single underlying mechanism uniting all Ga pathologies.

2. The concept of neomorphic mechanism as presented by the authors can't be agreed with. They propose that all mutations overload the Ric8 chaperone system and thereby cause Ga pathologies. This seems to be an gross over-generalization for many reasons. First, the assays discussed need not necessarily be incompatible as they are all in vastly different systems and measure different biological processes. The authors seem to be making assumptions that certain biochemical processes (e.g. signal transduction) must correspond with other processes (e.g. GTP uptake) and must then necessarily lead to certain conditions at the organismal level (e.g. larval/embryonic viability, clinical severity). Second, the authors fail to provide convincing evidence that overloading the Ric8 chaperone system is as widespread and harmful as they claim. Their evidence is based on only a handful of mutations from two papers, written by the authors themselves. Of the three GNAO1 hotspot mutations analyzed (accounting for roughly 30% of all GNAO1 mutations), only one (G203R) had increased interaction with Ric8B, with an accompanying correlation to disease onset. While all three had increased interactions with Ric8A, the correlation with disease onset was weak at best. Furthermore, additional analysis comparing Ric8 effects with disease severity is warranted beyond correlations with disease onset. Beyond this point, the "neomorphic" classification as defined by Muller is dubiously applied to current situation with Ga mutations. Ric8 binding is a part of the natural mechanism in ontogeny of Ga. Same as Gbg binding, same as GPCR binding. Some mutations exaggerate Ric8 association, true. But this would be expected from thermolabile, poorly folded proteins. Nucleotide mishandling causes this exact effect. However, not all, and in fact possibly even the majority of mutations found across Ga pathologies (especially cancer via GNAQ) are not thermolabile and thus would not be expected to sequester Ric8 proteins (Silvio Gutkind's work and work by Dohlman). Many mutations very similarly cause trapping of GPCRs and Gbg which received little attention.

3) From the arguments above it looks more likely that more nuanced multi dimensional model would be required to explain pathological mechanisms of Ga mutations. Such attempts have been made in the literature, e.g. see Knight et al 2023 but new mutations continue to arise and challenge straightforward classification schemes. The model proposed by the authors is heavily

inferred from mutations in Gao. The authors go on to suggest that not only could this homology model predict pathologies of potential mutations across Ga subtypes, but it could also be used to prescribe therapies for specific families of mutations. Thus, at this point, the authors pivot from an outlook centered on their proposed neomorphic mechanism to a more precision medicine-based approach. While such a multidimensional predictive model would be monumental in the field of Ga pathologies, it is unlikely that the model could be based on amino acid locations, particularly in Gao. Apart from conserved catalytic residues, each Ga subtype is specifically tailored for interactions with preferred receptors, effectors, and auxiliary proteins. The severity of mutations in these regions would be expected to be unique for each Ga subtype. Furthermore, Gao has no direct effectors and will therefore be underrepresented in mutations at the homologous loci needed for effector interactions. Lastly, there is evidence that mutations that affect nucleotide handling in Gao do not necessarily have severe clinical manifestations (Dominguez-Carral, 2023). The same cannot be assumed true for other Ga subtypes. These mutations would likewise be expected to be underrepresented.

4) While it is possible that a unifying drug discovery effort could be applied specifically towards the mishandling of nucleotide, this should not directly flow from the periodic table proposed here. The zinc treatments studied by the authors of this article are promising for GNAO1 conditions and might apply to other Ga subtypes due to similarities between the G protein nucleotide binding pockets. Other aspects of G protein signaling, such as receptor engagement, RGS engagement, misfolding problems, are unique to each Ga subtype. It is not obvious why mutations causing permanent effector binding or lack of Gbg release are universally addressable.

In conclusion, the authors have highlighted in great detail potential overarching dimensions of Ga pathologies, but their analysis regarding which dimensions are most important is still lacking. Tellingly, they have not produced a practical predictive model based on their analysis. It is unlikely that the field of Ga pathologies is in a state to make such a model. Rather, further biochemical and clinical evaluations of mutations in other Ga subtypes are needed before such a model can be completed. Thus, it is recommended that future revisions of the article emphasize the sections annotating and commenting on diverse Ga mutations and less on the evaluating which dimensions should be included in an over-arching model.

Dear Vladimir,

Thank you again for your patience while I sought advice on your article. As communicated previously, it has been difficult to find a suitable, unbiased expert, and I apologize for the delay this has caused in getting back to you.

However, we have finally found an advisor who has read your manuscript, the referees' comments, and your rebuttal. Given the conflicting reports we initially received, we specifically asked this expert to comment on the periodic table concept, bearing in mind the nature of our 'Perspective' articles.

This advisor has now responded, stating the following:

"I read the article and the rebuttal carefully and with great interest. The authors did an amazing job compiling literature on Ga disease mutants responsible for rare diseases. I am convinced that the G protein community and individuals interested in G protein-dependent disease mechanisms will appreciate the work. However, I regret to say that I share the critical reviewer's concerns; I had the same problem. I am hesitant and unconvinced by the periodic table metaphor, and I don't see why any link to the periodic system of chemical elements is necessary or helpful. I struggle with this link and get distracted by it when reading (it feels so oversold). I would expect some visualization of this link in a table. The authors call Figure 3 a table, which is, in my opinion, quite a stretch."

Based on the feedback from this advisor and referee #2, and after discussing your article within the team once more, we have decided to maintain our original decision and consider a revised article that avoids the periodic table concept.

I understand that you are keen to retain this concept, and we would of course respect your decision to submit your article elsewhere. In this case, we would appreciate a message to this effect.

I am very sorry that the outcome of the consultation is disappointing and would like to reassure you that we have made this decision after carefully considering all the feedback we received.

With kind regards,

Lise

Lise Roth, PhD

Senior Editor

EMBO Molecular Medicine

Referee #1 (Remarks for Author):

The review article by Katanaev and Solis discusses a very interesting topic. The periodic table of dominant Galpha mutations - is very exciting. It has a highly useful predictive power: it tells which mutations will be discovered in the future in the Galpha genes already known to cause disease, and even which new rare diseases will emerge in the future caused by mutations in the Galpha genes not previously associated to any pathogenic variants. This concept further predicts the unifying molecular mechanism of the diverse diseases caused by Galpha mutations. And finally, it predicts that the same drug can be proven applicable and useful across the spectrum of Galpha-pathies, despite these diverse diseases (developmental, oncogenic, metabolic, neurological, immune, etc.).

Expanding on the therapeutic side and including potential approaches other than small molecules would be helpful and informative. Although CRISPR and base editing are specific to some mutations, they could still be relevant to treating some of the diverse diseases. This addition and discussion would be highly recommended for publications in EMM.

We sincerely thank the Reviewer for the thoughtful, encouraging, and constructive comments on our manuscript. In response to the Reviewer's helpful suggestion to expand on therapeutic perspectives beyond small molecules, we have now included a dedicated section in the revised manuscript. This new text discusses the current treatment strategies as well as emerging gene-based strategies, including CRISPR-Cas9 and AAV-mediated RNA interference approaches, as potential avenues for treating selected G α -related disorders (first 2 paragraphs of the "Unified drug discovery for G α -pathies" section in p.10-11). We hope these additions address the Reviewer's request and further enrich the therapeutic dimension of the manuscript.

Once again, we deeply appreciate the Reviewer's support and insightful feedback.

Referee #2 (Remarks for Author):

The review by Katanaev and Solis summarizes the information about mutations across all Ga subtypes into a unified table of predictable mutations. The authors focus on mutation position, nucleotide exchange, hydrolysis, and gain/loss of function to argue mutations in disparate G alpha subunits have predictable biochemical and functional outcomes. The authors also focus on so-called neomorphic gain of function mutations which they themselves characterized in previous publications and define as increased binding to known chaperone proteins ric8a/b. Three main postulates are formed at the end of the article: 1) New mutations will be discovered at identical positions to those already characterized in

GNAO1 encephalopathy; 2) Nucleotide mishandling is a unifying molecular mechanism for Ga pathologies 3) A unified drug discovery paradigm can be expected whereby drugs effective for one Ga type can be applied to many more.

The authors go to great lengths to characterize mutations from multiple Ga subtypes that have arisen over decades of research and attempt to explain the wide variety of clinical and cellular phenotypes with the above postulates.

A comprehensive picture of diverse pathological Ga missense mutations is a significant addition to clinicians and G-protein field specialists interested in these rare diseases.

However, the article as presented has serious flaws.

We thank Reviewer #2 for the thorough review of our manuscript. We greatly appreciate his/her insightful comments and constructive feedback, which helped us to significantly improve the quality and clarity of the manuscript. We believe that the revised version fully addresses all of his/her criticisms and suggestions.

1. The concept of periodic table is grossly oversold. At its core, for anyone studying G proteins the concept of conserved structural elements in Ga subunits doing conserved things and their disruptions having easily predictable effects consistent with their functional role in G proteins (e.g. interaction with GPCRs, nucleotide binding and hydrolysis, association with Gbg, etc) conserved across different Ga isoforms is foundational. It is firmly in place based on results of many labs over decades and at this point rather trivial that does not require "Mendeleev" to put a stake in it. If you go beyond this concept - one very quickly gets into the fast sand and things become complicated beyond an ability to present a clear "periodic table". The principles for organizing such a table were not made clear by the authors and they actually do not include any table, beyond surface level common knowledge arguments.

We thank the Reviewer for this comment. To improve clarity and focus, we have removed the "Periodic Table" analogy from the manuscript. We agree that simplifying the narrative in this way helps ensure the message is more direct and accessible to the reader.

However, we would like to stress the central point of our Perspective, that is reflected already in the title: "*dominant Ga mutations in human disease...*". Sure enough, we are relying on the fundamental findings obtained by many labs (including our own) studying G proteins, but we focus on pathogenic mutations and are putting these findings in the framework of the diseases these mutations cause. Our analysis predicts that new mutations will be discovered in various rare diseases, predicts which mutations will produce more pronounced clinical manifestations, and even predicts which organs/tissues will be targeted by new rare diseases to be discovered in the future. We feel that this element is ignored by

the Reviewer, yet it is central to our Perspective. Finally, our model also predicts that drugs that are active against some G α -pathies, might be active against other, very distinct (in the clinical sense) G α -pathies.

Rather, the authors provide lists of overarching mechanisms, which themselves are poorly defined and inconsistent. Furthermore, competing language such as "unifying mechanism" and "multiparametric matrix" to describe the proposed periodic table highlight the paradoxical nature of making a model simultaneously reductive and nuanced. Therefore, is unclear whether the authors are ultimately arguing that Ga pathologies should be analyzed along several dimensions, or whether there is a single underlying mechanism uniting all Ga pathologies.

We thank the Reviewer for this important observation. In response, we have clarified throughout the revised manuscript that we are referring to multiple, convergent pathogenic mechanisms rather than a single one. We have made a concerted effort to consistently use language that reflects this plurality and to avoid terms that may imply an overly reductive model.

Yet, we are puzzled by the statement of the Reviewer that the mechanisms we described in our Perspective "*are poorly defined and inconsistent.*" Throughout the Perspective, we meticulously analyze all reported mechanisms, from one G α -subunit to the next, citing papers from multiple laboratories. There is a great deal of detail and consistency here.

2. The concept of neomorphic mechanism as presented by the authors can't be agreed with. They propose that all mutations overload the Ric8 chaperone system and thereby cause Ga pathologies. This seems to be an gross over-generalization for many reasons

We appreciate the Reviewer's engagement with the neomorphic aspect of the manuscript. However, we believe there may have been a misunderstanding, as we are not proposing that "*all mutations overload the Ric8 chaperone system*". As we explicitly described in that section of our Perspective (first paragraph in p.10) and also in Figure 3: "*GNAO1-mutations can be broadly grouped in three main categories (Fig. 3)...And Gao mutants carrying a near-normal Ric8 association define the third category (L) that contains partial LOF GNAO1-mutations leading to mild phenotypes such as late-onset dystonia and parkinsonism.*" We do make clear distinctions between the pathogenic variants that engage the Ric8 system (and lead to the more severe cases) and those that do not. And certainly, many non-pathogenic mutations can be found that do not interfere with the Ric8 system at all.

First, the assays discussed need not necessarily be incompatible as they are all in vastly different systems and measure different biological processes. The authors seem to be making assumptions that certain biochemical processes (e.g. signal transduction) must correspond with other processes (e.g. GTP uptake) and must then necessarily lead to certain conditions at the organismal level (e.g. larval/embryonic viability, clinical severity).

We are truly confused by these comments. As biologists, it is natural for us to hypothesize that the biochemical process a given protein participates in translates, to a certain degree, into the biological process that this protein regulates. In case of a pathogenic mutation in this protein translates into clinical manifestations or measurable defects in model organisms, is in fact, to a large extent, the basis for the existence and usage of model organisms.

Second, the authors fail to provide convincing evidence that overloading the Ric8 chaperone system is as widespread and harmful as they claim. Their evidence is based on only a handful of mutations from two papers, written by the authors themselves. Of the three GNAO1 hotspot mutations analyzed (accounting for roughly 30% of all GNAO1 mutations), only one (G203R) had increased interaction with Ric8B, with an accompanying correlation to disease onset. While all three had increased interactions with Ric8A, the correlation with disease onset was weak at best. Furthermore, additional analysis comparing Ric8 effects with disease severity is warranted beyond correlations with disease onset.

We respectfully note that several of the Reviewer's statements appear to be factually incorrect or based on a misinterpretation of the data. We would like to clarify these points below and explain the basis for our conclusions.

First, our analysis of GNAO1 variants does not deal with a "handful of mutations". Instead, the Ric8 dominance has been studied for the following pathogenic Gao variants: G40R, G45E, S47G, D174G, L199P, G203R, R209C, C215Y, A227V, Y231C, Q233P, E237K, E246K, N270H, F275S, I279N, Q52R (DOI: [10.1172/JCI172057](https://doi.org/10.1172/JCI172057)); L13P and L23P (DOI: [10.1002/mds.29720](https://doi.org/10.1002/mds.29720)); F251L and S264F (DOI: [10.1016/j.gendis.2025.101522](https://doi.org/10.1016/j.gendis.2025.101522)); C225Y and C225R (DOI: [10.1002/mco2.70196](https://doi.org/10.1002/mco2.70196)). This is hardly a "handful" of mutations (of note, the two last papers from this list have been published after the submission of this Perspective. Further, many more variants are routinely analyzed by us for the dominant Ric8A/B binding).

Second, the Reviewer is mistaken in his/her claims regarding the hotspot mutations. In our recent paper (DOI: [10.1172/JCI172057](https://doi.org/10.1172/JCI172057)), out the many mutations analyzed, 5 hotspot GNAO1 mutations were included. The gain of Ric8B binding for these hotspots, that cover

40% of all cases reported according to (DOI: [10.1155/2023/6628283](https://doi.org/10.1155/2023/6628283)), is as follows: G40R shows a 9.6-fold increase in Ric8B binding, G203R – 8.1-fold increase, E237K – 5.2-fold increase, E246K – 3.9-fold increase, and R209C – 2.8-fold increase. Each of these increases are indeed significant when compared individually to the wild-type Gao (see the presentation of the data in the preprint version of this article, DOI: [10.1101/2023.03.27.534359](https://doi.org/10.1101/2023.03.27.534359)). These highly reproducible findings permitted to calculate a strong correlation between the strength of Ric8B engagement and the clinical severity of the GNAO1 disease.

Third, we state as data not shown, that we observed the dominant Ric8A and Ric8B binding by pathogenic G α proteins other than Gao.

Forth, increased Ric8A binding was indeed previously reported for several pathogenic Gai3 mutants (DOI: [10.1126/scisignal.aad2429](https://doi.org/10.1126/scisignal.aad2429)), as we clearly described in our Perspective (second paragraph in p.10). The authors of that study, however, did not “connect the dots” and assumed this effect was related to the GEF activity of Ric8A instead.

Finally, the Reviewer’s comment regarding the disease onset as a marker for disease severity is puzzling. If we understand it right, the Reviewer questions whether the disease onset can be a valid meter of disease severity. Not only has this meter been well-accepted by the reviewers of the JCI article mentioned above (DOI: [10.1172/JCI172057](https://doi.org/10.1172/JCI172057)), but our numerous presentations of this work at clinical congresses and countless discussions with pediatric neurologists also confirm the validity of this approach.

Beyond this point, the "neomorphic" classification as defined by Muller is dubiously applied to current situation with G α mutations. Ric8 binding is a part of the natural mechanism in ontogeny of G α . Same as Gbg binding, same as GPCR binding.

Although the neomorphic nature of the severe pathogenic GNAO1 variants has been the subject of two experimental papers of ours (Science Advances 2022 and J Clin Invest 2024), and we do not see why this concept should be attacked in this Perspective, we respectfully suggest that the Reviewer may have misunderstood the concept of neomorphic mutations as it applies in this context. Hermann J. Muller, who got a Nobel prize for his foundational work on genetics, has defined the neomorphic mutation as follows: “neomorph represents a change in the nature of the gene... giving an effect not produced, or at least not produced to an appreciable extent, by the original normal gene”. In this regard, both the massive Ric8A binding by pathogenic Gao (with wild-type Gao only weakly binding Ric8A), and even more so the strong trapping of Ric8B (as Ric8B is ‘foreign’ for most G α -subunits, dedicated to Gas/Gaolf instead) clearly fall under the Muller neomorph definition.

Some mutations exaggerate Ric8 association, true. But this would be expected from thermolabile, poorly folded proteins. Nucleotide mishandling causes this exact effect. However, not all, and in fact possibly even the majority of mutations found across Ga pathologies (especially cancer via GNAQ) are not thermolabile and thus would not be expected to sequester Ric8 proteins (Silvio Gutkind's work and work by Dohlman).

We appreciate the Reviewer's perspective; however, the suggestion that thermolability underlies the gain of neomorphic Ric8 binding by GNAO1 mutations is not currently supported by experimental evidence. In fact, a recent study by the Dohlman's group (DOI: [10.1016/j.celrep.2023.113462](https://doi.org/10.1016/j.celrep.2023.113462)) analyzed many pathogenic GNAO1 mutations, and found that some mutants (called group 2 in the study) show near-normal thermolability in the GDP-loaded state and only a small reduction in stability as GTPγS-loaded. Although the authors did not run a statistical analysis to confirm the significance of their data, no direct correlation between thermolability and sequestering Ric8 can be inferred from our analysis of the same mutants. For example, the hotspot G203R and R209C mutants show a very similar thermolability, but different Ric8B binding (see above). Additionally, even if the neomorphic Ric8 binding of Gα mutants might be "obvious", the global implications that sequestering these chaperons might have on the entire G proteins network was completely ignored until now.

The cancer-causing GNAQ mutations the Reviewer mentions (such as Gαq [Q209P] in uveal melanoma) are irrelevant to the topic of our Perspective. These mutations are the classical constitutively active mutants, "*have historically dominated the field of disease-related heterotrimeric G protein research*" as we write in the beginning of our article (third paragraph in p.2), and have a completely different mechanism of pathogenicity than the pathogenic mutations our Perspective is dedicated to.

Many mutations very similarly cause trapping of GPCRs and Gβγ which received little attention.

It may be that the Reviewer overlooked the section in which we specifically address this point. In fact, we dedicate an entire paragraph to discussing the available data on pathogenic Gα mutants that trap GPCRs (last paragraph in p.8), beginning with the sentence: "*Probably one of the most striking aspects of several GNAO1-mutations is that they lead to a dominant GPCR-coupling.*"

We acknowledge, however, that our original version did not explicitly discuss the trapping of G $\beta\gamma$. In this revised version, we have added a couple of sentences summarizing the available data on G α o mutants that appear to sequester G $\beta\gamma$ (first paragraph of the “Functional consequences of dominant GNA mutations” section; p.7). To the best of our knowledge, the sequestration of G $\beta\gamma$ by other pathogenic G α variants has not yet been reported.

3) From the arguments above it looks more likely that more nuanced multi dimensional model would be required to explain pathological mechanisms of G α mutations. Such attempts have been made in the literature, e.g. see Knight et al 2023 but new mutations continue to arise and challenge straightforward classification schemes. The model proposed by the authors is heavily inferred from mutations in G α o.

We certainly discuss in a great detail the data by Knight et al. (dedicated also to G α o), as well as from the other relevant papers on G α o and other G α -subunits. As commented above, in this new revised version we have clarified the multidimensionality of the mechanisms behind pathogenic mutations in G α -subunits.

The authors go on to suggest that not only could this homology model predict pathologies of potential mutations across G α subtypes, but it could also be used to prescribe therapies for specific families of mutations. Thus, at this point, the authors pivot from an outlook centered on their proposed neomorphic mechanism to a more precision medicine-based approach. While such a multidimensional predictive model would be monumental in the field of G α pathologies, it is unlikely that the model could be based on amino acid locations, particularly in G α o. Apart from conserved catalytic residues, each G α subtype is specifically tailored for interactions with preferred receptors, effectors, and auxiliary proteins. The severity of mutations in these regions would be expected to be unique for each G α subtype. Furthermore, G α o has no direct effectors and will therefore be underrepresented in mutations at the homologous loci needed for effector interactions.

Our Perspective is based on the premise that mutations in conserved residues might and probably will emerge for additional G α -subunits in connection with genetic disorders. This is based on the shared mechanisms among pathogenic G α mutations reported in the literature, plus the neomorphic feature we uncover for GNAO1. We agree with the Reviewer that G α -subunits have non-conserved regions that are unique for each G α -subfamily, and mutations in those regions might not be predicted by our analysis. As most unique regions among G α

subfamilies reside within the alpha-helical domains, we exclude this region in our alignment in Fig. 3. Thus, we focused mostly on conserved regions, as we speculate that mutations in the non-conserved regions might lead to milder phenotypes among the G α -pathies, or might not even lead to any phenotype at all.

Lastly, there is evidence that mutations that affect nucleotide handling in Gao do not necessarily have severe clinical manifestations (Dominguez-Carral, 2023). The same cannot be assumed true for other Ga subtypes. These mutations would likewise be expected to be underrepresented.

It appears there may be a misunderstanding regarding the findings of Dominguez-Carral, 2023. The study did not analyze the biochemical properties of the 12 GNAO1 mutations studied (involving 16 patients), and therefore nucleotide handling defects cannot be directly inferred. Also, 4 of the 12 GNAO1 mutations had not been reported elsewhere. Additionally, authors described as “mild” patients with mutations in the hotspot R209, which is known to lead to the NEDIM phenotype. Patients on the milder spectrum of GNAO1-disorders – such as adolescent/adult-onset dystonia, parkinsonism, and autism with intellectual disability – are outside of the main focus of this study.

4) While it is possible that a unifying drug discovery effort could be applied specifically towards the mishandling of nucleotide, this should not directly flow from the periodic table proposed here. The zinc treatments studied by the authors of this article are promising for GNAO1 conditions and might apply to other Ga subtypes due to similarities between the G protein nucleotide binding pockets. Other aspects of G protein signaling, such as receptor engagement, RGS engagement, misfolding problems, are unique to each Ga subtype. It is not obvious why mutations causing permanent effector binding or lack of G $\beta\gamma$ release are universally addressable.

We appreciate the fact the Reviewer, despite his/her criticisms above, does agree with this key element of our Perspective. As we write as data not shown, we already have the preliminary data that several pathogenic G α i2 variants that lead to rare yet devastating immune dysfunctions (Science 2024, DOI: [10.1126/science.add8947](https://doi.org/10.1126/science.add8947)) do biochemically respond to zinc in the manner similar to that of pathogenic Gao, opening truly exciting treatment opportunities. How many other pathogenic mutations across different G α -pathies can be universally addressed with the same pharmacological approaches will be determined by future experiments. The goal of our Perspective is to trigger such future experimentation.

In conclusion, the authors have highlighted in great detail potential overarching dimensions of Ga pathologies, but their analysis regarding which dimensions are most important is still lacking. Tellingly, they have not produced a practical predictive model based on their analysis. It is unlikely that the field of Ga pathologies is in a state to make such a model. Rather, further biochemical and clinical evaluations of mutations in other Ga subtypes are needed before such a model can be completed. Thus, it is recommended that future revisions of the article emphasize the sections annotating and commenting on diverse Ga mutations and less on the evaluating which dimensions should be included in an overarching model.

It appears there may be a misunderstanding regarding the scope and purpose of our manuscript. This is what our Perspective already contains: a meticulous analysis of the biochemical and cellular deficiencies of pathogenic G α variants across G α -subunits. As we are presenting a Perspective, and not a review nor experimental paper, the expectation that we would include “further biochemical and clinical evaluations” seems somewhat misplaced.

27th Jun 2025

Dear Dr. Katanaev,

Thank you for submitting your revised manuscript to EMBO Molecular Medicine, which has been reviewed by the expert who had advised us on the initial submission (now referee #3). As you will see below, this referee is satisfied with the revisions, and I will thus be able to accept your manuscript once the following points are addressed:

1. While we do not have strict limits in the number of references, we recommend a maximum of 50 references, much less than your current 131 references. Could you please reduce the total number of references?
2. For the figures, please note the following:
 - If there are certain aspects of your figure draft that are based upon assumptions or where the scientific data remains ambiguous, please add a comment so that we can work with you on an accurate depiction. Please ensure the directionality and nature of interactions is presented accurately.
 - If the figure or single panels of the figure have been adapted from a published figure, please add this information to the figure legend (e.g., 'Adapted from...' or 'Based on...').
 - Please only re-use figures or parts of a figure if this is essential for understanding the concept communicated. If the figure contains re-used images or elements of images, please make sure that you have the permission/license to publish it (this also applies to your own previous work, if the journal you published in retains copyright). All re-used material must be explicitly cited.
 - If you use an image data base for scientific iconography (e.g., BioRender), please let us know if you have a license that allows for publication in an academic journal. Please ensure the information shown is scientifically accurate. If in doubt, please discuss with the editor or provide a sketch so that our designers can create accurate iconography.

In particular:

Figure 1: as suggested by the referee, please consider changing the green arrow design.

Figure 2B: please clarify the source/methodology.

Figure 2C: please clarify the source/methodology.

Figure 3: please indicate the source of the data.

I look forward to receiving your revised manuscript.

Yours sincerely,

Lise Roth

***** Reviewer's comments *****

Referee #3 (Remarks for Author):

I appreciate the changes made by the authors, particularly the removal of the rather confusing periodic table of elements message. As it stands, I see this piece of literature as an informative and comprehensive overview of diverse pathological Ga missense mutations in rare diseases. I do not think that any additional changes should be made, except for some visual improvements to Fig. 1: the green arrow is oddly shaped and does not fit with the rest of the cartoon.

The authors addressed the remaining editorial issues.

4th Jul 2025

Dear Dr. Katanaev,

We are pleased to inform you that your manuscript is accepted for publication and is now being sent to our publisher to be included in the next available issue of EMBO Molecular Medicine.

Your manuscript will be processed for publication by EMBO Press. It will be copy edited and you will receive page proofs prior to publication.

This Perspective is free of charge, and we will shortly send you an email with a token. When you are contacted in a few weeks to sign your license agreement and review article proofs, please enter this token into the relevant field in the Springer Nature author services system.

With kind regards,

Lise
